# Hepatic Epithelioid Hemangioendothelioma in a Dog

**DOI:** 10.3390/ani14091302

**Published:** 2024-04-25

**Authors:** Luisa Vera Muscatello, Federico Massari, Paola Roccabianca, Giuseppe Sarli, Cinzia Benazzi, Marco Luigi Bianchi

**Affiliations:** 1Department of Veterinary Medical Sciences, University of Bologna, Via Tolara di Sopra, 50, Ozzano dell’Emilia, 40064 Bologna, Italy; giuseppe.sarli@unibo.it (G.S.); cinzia.benazzi@unibo.it (C.B.); 2DOCVET Clinica Veterinaria Nervianese, Via Rho 2, 20014 Nerviano, Italy; massari@docvet.it (F.M.); marcoluigi.bianchi@gmail.com (M.L.B.); 3Department of Veterinary Medicine, University of Milano, via dell’ Università 6, 26900 Lodi, Italy; paola.roccabianca@unimi.it

**Keywords:** liver, epithelioid hemangioendothelioma, dog, immunohistochemistry

## Abstract

**Simple Summary:**

Hemangioendothelioma is an uncommon neoplasm that affects both humans and animals. Its clinical course is unclear. This report describes the first case of hepatic epithelioid hemangioendothelioma in a dog. The clinical and microscopic characteristics of this neoplasm overlap with those described in humans; therefore, this report can be informative as a spontaneous animal model but, at the same time, also useful for veterinary progress in diagnostic orientation and therapeutic treatment.

**Abstract:**

A 5-year-old spayed female Breton dog was referred for a thyroid nodule. A total body CT scan evidenced multifocal hepatic nodules. Cytological liver samples were hemodiluted and non-diagnostic. Following a thyroidectomy, the histology was consistent with a follicular-compact thyroid carcinoma. On laparoscopy, most hepatic lobes had multifocal dark-red nodules that were biopsied for histology. Microscopically, the hepatic parenchyma in the nodules was substituted by blood channels lined by bland spindle cells but adjacent to epithelioid neoplastic cells, single or in clusters, embedded in a moderate amount of edematous collagen matrix. These cells had optically empty cytoplasmic space, occasionally containing erythrocytes (microlumina). Spindle and epithelioid cells expressed membranous-to-cytoplasmic CD31 and FVIII-RA consistent with endothelial origin. Based on morphology and immunolabelling, a hemangioendothelioma with epithelioid differentiation was diagnosed. Lesions in the liver were initially stable, showing progression with time. The dog was alive with no systemic clinical signs 36 months after laparoscopy.

## 1. Introduction

Proliferative vascular disorders comprise malformations, reactive proliferations, and vascular neoplasms [1]. Vascular malformations are defined as anomalies in angioarchitecture, such as arteriovenous malformations, telangiectasis, reactive angiomatosis, and vascular hamartoma [1]. Other rare vascular conditions considered reactive include feline reactive angioendotheliomatosis and papillary endothelial hyperplasia. Feline reactive angioendotheliomatosis is a multisystem disease characterized by whorled glomeruloid endovascular proliferations composed of endothelial cells and lesser pericytes [1,2].

Papillary endothelial hyperplasia is a reactive vascular disorder with a complex classification and pathogenesis, but generally considered an excessive reaction to thrombosis, that has been described in dogs and cats [1,3].

The spectrum of blood vessel neoplasms comprises benign (hemangioma), intermediate (hemangioendothelioma), and malignant (hemangiosarcoma) tumors. Of these, hemangioendothelioma seems rare in most species and has been occasionally reported in the skin, spleen, lung, and brain of dogs [4,5,6,7,8,9]. Different microscopical variants have been reported, including epithelioid, retiform, and kaposiform types [1,6,7,9].

Primary epithelioid hemangioendothelioma of the liver is an uncommon tumor described in humans, with an incidence of less than 0.1 per 100,000 population [10,11]. Hepatic epithelioid hemangioendothelioma occurs most frequently in middle-aged women and it is grossly classified into solitary, multiple, or diffused forms [12].

Due to the rarity of this tumor, the outcome and the molecular background are still poorly characterized [11]. Errani et al. [13] found that WWTR1-CAMTA1 was a recurrent mutation in human epithelioid hemangioendothelioma regardless of the tumor site. The involved genes function as transcription co-activators and tumor suppressor genes, and their mutations are thought to result in oncogenesis [13].

The aim of this report is to describe the clinical presentation, histopathology, and biological behavior of canine hepatic hemangioendothelioma.

## 2. Materials and Methods

A 5-year-old spayed female Breton dog was referred for a left thyroid nodule. The dog was clinically examined and investigated using computed tomography (CT) scan and ultrasound. CT scans were performed on the skull, cervical region, thorax, and abdomen pre- and post-IV administration of non-ionic iodate contrast agent. No triple-phase scan was performed, and CT was acquired using 2 mm slice pre- and post-contrast. Thyroidectomy and abdominal laparoscopy with hepatic biopsies were performed. Procedure was performed with two 5 mm ports inserted in the ventral midline. Dog was tilted on both sides to fully explore the liver, and, with a 5 mm cup, hepatic tissue was collected. No bleeding was elicited during the procedure.

Tissue samples were fixed in buffered formalin and routinely processed and stained with hematoxylin and eosin.

Serial 3 μm-thick sections were obtained and placed onto glued slides for immunohistochemistry using antibodies against CD31 (clone IC70A, cod. M0823, Dako Glostrup, Denmark; dilution 1:30) and against FVIII-RA (polyclonal, cod. A0082, Dako Glostrup, Denmark; dilution 1:2000). Antigen retrieval was performed, for CD31, via incubation in EDTA buffer, pH 8.0, for 10 min, in a microwave oven at 750 W, followed by enzymatic retrieval with Pepsine 0.05%, pH 7.5, for 15 min in oven at 37 °C, while sections for FVIII-RA were retrieved in citrate buffer, pH 6.0, for 10 min in a microwave oven at 750 W. Binding sites were revealed via secondary biotinylated antibody (dilution 1:200) and amplified using a commercial avidin-biotin-peroxidase kit (VECTASTAIN; ABC Kits, Peterborough, UK). The chromogen 3,3′-diaminobenzidine was used. Slides were counterstained with Harris hematoxylin. Positive control tissue consisted of canine skin with granulation tissue. For the negative controls, the primary antibody was omitted.

The patient was monitored every two months, and follow ups were performed with clinical examination and three consecutive ultrasound assessments during the first 6 months after laparoscopy.

## 3. Results

### 3.1. Clinical Findings

The clinical examination of the patient was normal. Abdominal palpation was unremarkable. The preoperative total-body CT-scan detected a space-occupying lesion with net margins, an oval morphology, and a volume equal to 3.8 × 3 × 2.4 cm at the left thyroid lobe. Such an imprint lesion, without a sharp plane of adipose cleavage on the adjacent portions of the trachea, appeared iso-hypoattenuating to the cervical musculature on the basal scan with heterogeneous enhancement in the post-contrast scan. There was no evidence of vascular invasion. Furthermore, via CT-scan, we detected multifocal, variably sized, hepatic nodules (Figure 1a), with the largest-sized being 4.6 cm in the caudate process of the caudate lobe, iso-attenuating compared to the hepatic parenchyma in basal-scan conditions, with weak enhancement in post-contrast scanning (Figure 1b).

Ultrasound-guided fine-needle aspiration cytology of the liver lesions yielded poorly cellular, hematic, non-diagnostic samples.

On laparoscopy, the hepatic lesions were multifocal, dark-red, intraparenchymal nodules involving several hepatic lobes (Figure 1c,d).

### 3.2. Histopathological and Immunohistochemical Features

The thyroid nodule was consistent with a follicular-compact thyroid carcinoma with infiltrative capsular invasion and intravascular tumor emboli.

The hepatic parenchyma was expanded by a multifocal, poorly demarcated, unencapsulated proliferation composed of hematic lacunae (Figure 2a) lined by a monolayer of bland spindle cells, interpreted as ectatic and hyperemic vessels. Vascular luminal thrombosis was focally observed. Adjacent to the ectatic vascular structures there were clusters or single, polygonal epithelioid cells, embedded in a moderate amount of myxoid–edematous matrix (Figure 2a,b). The cells were of 20–30 microns, with variably distinct cells borders, an intermediate N/C ratio, and a moderate amount of diaphanous cytoplasm, often with a large single vacuole, displacing the nucleus at the cell periphery (signet-ring-like) and occasionally containing erythrocytes (microlumen formation). The nuclei were oval to flat. Anisocytosis and anisokaryosis were moderate, and mitoses were not observed (Figure 2b).

Signet-ring and microlumen-forming cells, along with cells bordering vascular lacunae, were diffusely positive, with intense membranous immunolabeling for CD31 (Figure 2c) and intense cytoplasmic-to-membranous immunolabeling for FVIII-RA (Figure 2d).

Overall, a diagnosis of hepatic epithelioid hemangioendothelioma was obtained.

### 3.3. Follow Up

The patient was monitored every two months without any medical therapy. Ultrasounds assessed stable liver lesions during the first 6 months after laparoscopy, but, subsequently, hepatic nodules progressively increased in size and stabilized again, as observed at the 20-month follow up. During this time, routine blood analysis and the clinical conditions of the dog were stable. An ultrasound at 27 months demonstrated further disease progression (Appendix A); there was an increase in the number and, above all, the size of the lesions already present. At this point, the dog was started on thalidomide, an antiangiogenic compound at 8 mg/kg every day, paralleling what was reported in human medicine [14].

The dog did not develop clinical signs, and all the tests were found to be normal (Appendix A).

## 4. Discussion

According to the human WHO classification [15], hepatic hemangioendothelioma is a malignant endothelial neoplasm composed of epithelioid cells embedded in a myxohyaline or fibrous stroma, with frequent microlumen formation. The microscopical features of the case in this dog paralleled those described in human patients [10,15].

Moreover, the neoplastic cells, in our case, often had a large cytoplasmic vacuole, occasionally containing erythrocytes, and were immersed in myxoid matrix. This morphology was interpreted as representing microlumen formation, a feature typical of hemangioendothelioma that was confirmed by the finding of intraluminal erythrocytes and by positive immunolabeling for the endothelial markers, CD31 and FVIII-RA, as described in humans where cells are labeled with the endothelial markers CD31, CD34, FVIII-RA, podoplanin, and ERG [10,15].

In humans, differential diagnoses of hepatic hemangioendothelioma include cholangiocarcinoma, metastatic signet-ring cell carcinoma, and the sclerosing variant of hepatocellular carcinoma [11,16]. Immunohistochemistry is useful for the diagnostic differentiation and to confirm the endothelial origin for hemangioendothelioma. Additionally, in humans, epithelioid hemangioendothelioma may be misdiagnosed as angiosarcoma, but the typical morphologic features should direct the pathologist to hypothesize hemangioendothelioma and to the correct diagnosis, as angiosarcoma has severe atypia, nuclear pleomorphism, and high mitotic activity contrary to hemangioendothelioma [11]. Therefore, even in dogs, immunohistochemistry is useful only to confirm the endothelial origin of the neoplasm. To differentiate a hemangioendothelioma from a hemangiosarcoma, a careful assessment of the histological features is necessary. In fact, in this case, the features of cellular atypia, nuclear pleomorphism, and mitotic activity were minimal and did not resemble the malignant morphological features typical of canine hemangiosarcoma.

In the case described, neoplastic cell aggregates were arranged adjacently to markedly ectatic vessels that lacked linin cell atypia and that were, thus, considered reactive and non-neoplastic. The ectasia of the lacuna-forming vessels can be induced, similarly to in humans, for hemangioendothelioma because of the compression and occlusion of vascular channels by neoplastic cells [10]. Occlusion resulting in vascular ectasia may be associated with thrombosis, a feature observed in this case and also reported for cases of human hepatic epithelioid hemangioendothelioma [10].

Histology is also essential in the diagnosis of hepatic hemangioendothelioma in humans, since, although a CT scan may have some typical patterns, such as “halo sign” and “lollipop sign” [8], imaging features are not sufficiently specific and metastatic disease or primary hepatic neoplasms with intrahepatic spreading cannot be ruled out on imaging [12]. Indeed, in our case, there were no specific imaging features, and, in differential diagnosis, a metastatic neoplasm (hypothetically from the thyroid carcinoma, as there was vascular embolization of cells) or a primary multifocal hepatic neoplasm were considered.

The hypoattenuation in the post-contrast scan can be a sign of malignancy, as already described in the veterinary literature [17]. This cannot be considered a pathognomonic feature, and, even if it is sometimes indicative of a malignant process, biopsy is usually recommended to define the nature of the lesion.

Multiple malignant neoplasms can occur in dogs, regardless of breed or sex predisposition. An interesting finding was reported in one study: 33% of dogs affected by thyroid carcinoma, as in the present case, have additional neoplasms [18]. It is therefore essential to always perform complete staging.

In humans, hemangioendothelioma has variable survival, with an indolent-to-progressive, and, rarely, metastatic behavior [10,15]. Considering these unpredictable behaviors, there are no standard guidelines for the prognostication and treatment of human hepatic epithelioid hemangioendothelioma [11]; the treatment options are extensive and have included chemotherapy, surgery, and liver transplantation [11,14]. However, there is a general consensus to start with clinical observation alone to evaluate the behavior of the tumor prior to more aggressive therapies, because, if the tumor is not aggressive, the spontaneous regression or stabilization of the disease has been reported, and overtreatment should be avoided [11].

In this dog, the tumor had a stable course for 6 months, with a subsequent increase in size of the lesions (between 6 and 27 months), followed by disease stabilization. At 36 months the patient was asymptomatic and in good clinical conditions, similarly to what has been reported in some cases in human medicine [11].

## 5. Conclusions

In conclusion, this report describes the first case of hepatic hemangioendothelioma in a dog. The tumor in the present report showed long periods of stable disease followed by a slow and asymptomatic progression during the 36th month of follow up.

## Figures and Tables

**Figure 1 animals-14-01302-f001:**
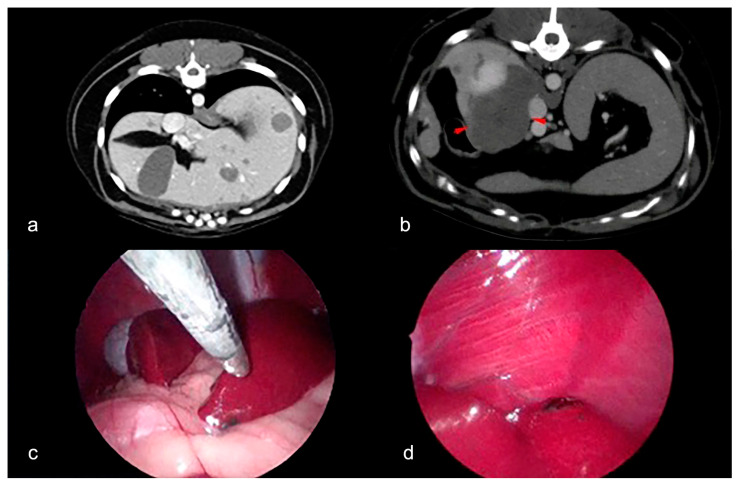
Epithelioid hemangioendothelioma in the liver of a dog. (**a**) CT-scan. Multifocal hepatic iso-attenuating liver lesions in basal scan with weak enhancement in post-contrast scanning. (**b**) CT-scan. The largest hepatic lesion (red arrowheads; caudate process of the caudate lobe). (**c**) Laparoscopy: 5 mm cup forceps collecting liver biopsy. Red lesion is also evident at the lobe margin. (**d**) Laparoscopy: liver lesions observed during laparoscopic exploration to perform biopsies.

**Figure 2 animals-14-01302-f002:**
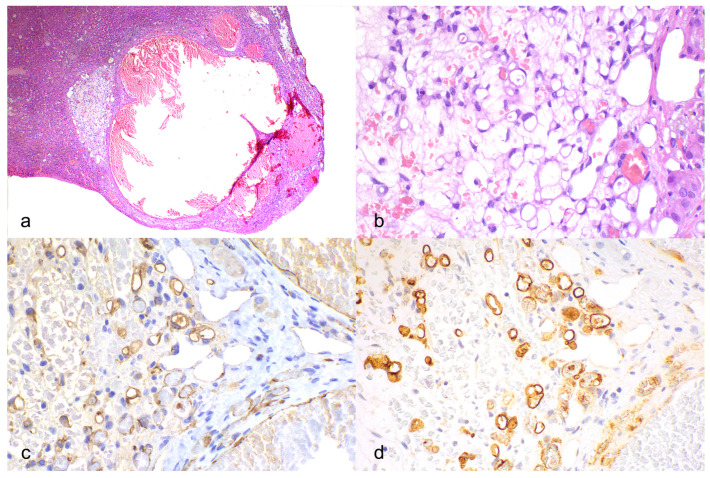
Microscopic features of epithelioid hemangioendothelioma in the liver of a dog. (**a**) Ectatic vascular lacunae with adjacent cluster of cells embedded in a prominent whitish myxoid matrix. Hematoxylin and eosin (HE). (**b**) High power of epithelioid cells with diaphanous cytoplasm and peripheralized moderately anisokaryotic nuclei (signet-ring-like); the neoplastic cells occasionally contain erythrocytes (microlumen formation). HE. (**c**) Immunohistochemistry for CD31. Neoplastic cells and vascular lacuna-lining cells were immunolabelled with intense membranous expression. (**d**) Immunohistochemistry for FVIII. Neoplastic cells and vascular lacuna-lining cells were immunolabelled with intense membranous-to-cytoplasmic expression.

## Data Availability

The original contributions presented in the study are included in the article/Appendix A; further inquiries can be directed to the corresponding author.

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
