# Peer review of "Hepatic Epithelioid Hemangioendothelioma in a Dog"

_animals, 2024, doi:10.3390/ani14091302_

Round 1

Reviewer 1 Report

Comments and Suggestions for Authors

This is an interesting case report about a relative young dog that  simultaneously developed a thyroid carcinoma and liver hemangioendothelioma. Nevertheless, tt seems that this paper would be more suitable for a short communication, rather than a full article. Plus I recommend some improvement on the discussion. There is nothing about differentiation of hemangioendothelioma from haemangiosarcoma, which is positive for the same markers used on immunohistochemistry but it would definetely carry a much worse prognosis. Also some discussion about the diagnosis of two different neoplasms in a young pure breed dog.

Author Response

Since hemangioendothelioma and hemangiosarcoma are both of endothelial origin, we mentioned in the discussion (lines 199-203) that the differentiation between the two entities is purely based on histological features. The discussion is as follows: “Additionally, in humans, epithelioid hemangioendothelioma may be misdiagnosed as angiosarcoma, but the typical morphologic features should direct the pathologist to hypothesize hemangioendothelioma and to the correct diagnosis as angiosarcoma has severe atypia, nuclear pleomorphism and high mitotic activity contrary to hemangioendothelioma.”

Furthermore, we added the following sentence (lines 203-208):

“Therefore, even in dogs, immunohistochemistry is useful only to confirm the endothelial origin of the neoplasm. To differentiate a hemangioendothelioma from a hemangiosarcoma, careful assessment of the histological features it is necessary. In fact, in this case, features of cellular atypia, nuclear pleomorphism and mitotic activity were minimal and did not resemble malignant morphological features typical of canine hemangiosarcoma.”.

Regarding your suggestion to discuss the presence of two tumors in a pure breed dog, we have added the following paragraph to the discussion (lines 227-230):

“Multiple malignant neoplasms can occur in dogs regardless of breed or sex predisposition. An interesting finding is that it was reported in one study that 33% of dogs affected by thyroid carcinoma, as in the present case, have additional neoplasms. It is therefore essential to always perform complete staging.”

Furthermore, considering that the manuscript describes a single case we considered the format for a case report, which is therefore a short communication.

Reviewer 2 Report

Comments and Suggestions for Authors

Dear Authors,

I reviewed the manuscript entitled "Hepatic epithelioid hemangioendothelioma in a dog". The report describes a case of a rare hepatic tumor in a dog with concurrent thyroid carcinoma, with clinical, diagnostic and laparoscopic findings, with a long-term follow-up. The topic is interesting, since is the first published case of this rare tumor in a dog.

I have only some minor comments (see below).

Specific comments:

Line 58: please switch to "the aim is to describe the clinical presentation, histopathology and biological behavior of canine hepatic hemangioendothelioma.

line 63: please expand CT, should be Computed Tomography (CT) scan.

Did the dog undergo CT first and then ultrasound? Were the authors aware of the presence of liver nodules before CT scan? 

Please explain CT protocol was used. Did the authors plan triple-phase scan?

Results

please describe more thoroughly liver nodule enhancement, since in humans hepatic hemangioendothelioma has peculiar pattern such as target or lollipop sign, as reported in the discussion section. Add a figure with a basal condition image, arterial phase, portal venous phase and late phase, if available. This can be interesting to the reader.

Please, add a brief description of the thyroid nodule (dimensions, side, CT and/or US features)

Line 177: hypoattenuation in portal/late phase is a feature of malignant hepatic lesions in dog (Griebie et al 2017). In fig 1 hepatic lesions appeared ad hypoattenuating after contrast medium injection. Please, discuss this feature more thoroughly.

Author Response

REVIEWER 2

Dear Authors,

I reviewed the manuscript entitled "Hepatic epithelioid hemangioendothelioma in a dog". The report describes a case of a rare hepatic tumor in a dog with concurrent thyroid carcinoma, with clinical, diagnostic and laparoscopic findings, with a long-term follow-up. The topic is interesting, since is the first published case of this rare tumor in a dog.

I have only some minor comments (see below).

Authors:

Dear reviewer,

we are grateful for your observations, and we have appreciated the criticisms that have helped to improve the manuscript. Please find below our responses

Reviewer 2: Line 58: please switch to "the aim is to describe the clinical presentation, histopathology and biological behavior of canine hepatic hemangioendothelioma.”

Authors: done.

Reviewer 2: line 63: please expand CT, should be Computed Tomography (CT) scan.

Authors: done

Reviewer 2: Did the dog undergo CT first and then ultrasound? Were the authors aware of the presence of liver nodules before CT scan? 

Authors: CT was performed prior to ultrasound for staging thyroid neoplasia.

Reviewer 2: Please explain CT protocol was used. Did the authors plan triple-phase scan?

Authors: We added the following sentence in the material and method (lines 84-86):

“CT of skull, cervical region, thorax, and abdomen pre and post IV administration of non-ionic iodate contrast agent was made. No triple-phase scan was performed, and CT was acquired using 2 mm slice pre and post contrast.”

Reviewer 2: Results

please describe more thoroughly liver nodule enhancement, since in humans hepatic hemangioendothelioma has peculiar pattern such as target or lollipop sign, as reported in the discussion section. Add a figure with a basal condition image, arterial phase, portal venous phase and late phase, if available. This can be interesting to the reader.

Authors:    As discussed in lines 216-222, there were no peculiar pattern in the present case: “although CT scan may have some typical patterns such as "halo sign" and "lollipop sign" [15], imaging features are not sufficiently specific and metastatic disease or primary hepatic neoplasms with intrahepatic spreading cannot be ruled out on imaging [5]. Indeed, in our case there were no specific imaging features and in differential diagnosis a metastatic neoplasm (hypothetically from the thyroid carcinoma as there was vascular embolization of cells) or a primary multifocal hepatic neoplasm were considered.”

No triple-phase scan was performed, CT was acquired using 2 mm slice pre and post contrast.

Reviewer 2: Please, add a brief description of the thyroid nodule (dimensions, side, CT and/or US features)

Authors: We added the following description (lines 112-116): Preoperative total body CT-scan detected space-occupying lesion at net margins, oval morphology, and volume equal to 3.8 x 3 x 2.4 cm at left thyroid lobe. Such imprint lesion with not sharp plane of adipose cleavage on the adjacent portions of the trachea, appears iso-hypoattenuating to the cervical musculature in basal scan with heterogeneous enhancement in post-contrast scan. No evidence of vascular invasion.

Reviewer 2: Line 177: hypoattenuation in portal/late phase is a feature of malignant hepatic lesions in dog (Griebie et al 2017). In fig 1 hepatic lesions appeared ad hypoattenuating after contrast medium injection. Please, discuss this feature more thoroughly.

Authors:  we added the following sentence in the discussion: “The hypoattenuation in the post contrast scan, can be a sign of malignancy as already described in veterinary literature. This cannot anyway be considered a pathognomonic feature, and even if sometimes indicative of a malignant process, biopsy is usually recommended to define the nature of the lesion.”

Reviewer 3 Report

Comments and Suggestions for Authors

The authors submit a case report to describe the first case of an " Hepatic epithelioid hemangioendothelioma in a dog".

 I really enjoy to red the manuscript. The structure of the manuscript is well organised. References are adequate and actual. 

Discussion is well structure and solid with and important comparative aspect.

Congratulations!

This case report has quality to be accepted. 

Lines 83-85 "The patient was monitored every two months and follow ups were performed by clinical examination and three consecutive ultrasound assessments during the first 6 months after laparoscopy. "

Lines 104-105 : The thyroid nodule was consistent with a follicular-compact thyroid carcinoma with infiltrative capsular invasion and intravascular tumor emboli. 105

Author Response

Dear reviewer,

we are grateful for the reviewing of our work and appreciation of the manuscript. Please find below our responses.

Reviewer 3: Lines 83-85 "The patient was monitored every two months and follow ups were performed by clinical examination and three consecutive ultrasound assessments during the first 6 months after laparoscopy. "

Authors: done.

Reviewer 3: Lines 104-105: The thyroid nodule was consistent with a follicular-compact thyroid carcinoma with infiltrative capsular invasion and intravascular tumor emboli. 105

Authors: done.